# Nitric Oxide Metabolism Affects Germination in *Botrytis*
*cinerea* and Is Connected to Nitrate Assimilation

**DOI:** 10.3390/jof8070699

**Published:** 2022-07-01

**Authors:** Francisco Anta-Fernández, Daniela Santander-Gordón, Sioly Becerra, Rodrigo Santamaría, José María Díaz-Mínguez, Ernesto Pérez Benito

**Affiliations:** 1Institute for Agribiotechnology Research (CIALE), Department of Microbiology and Genetics, University of Salamanca, 37008 Salamanca, Spain; u86028@usal.es (F.A.-F.); sioly@usal.es (S.B.); josediaz@usal.es (J.M.D.-M.); 2Facultad de Ingeniería y Ciencias Aplicadas (FICA), Carrera de Ingeniería en Biotecnología, Universidad de las Américas (UDLA), Quito 170513, Ecuador; daniela.santander.gordon@udla.edu.ec; 3Department of Computer Science, University of Salamanca, 37008 Salamanca, Spain; rodri@usal.es

**Keywords:** flavohemoglobin, development, nitrosative stress, secondary metabolism, transcriptome

## Abstract

Nitric oxide regulates numerous physiological processes in species from all taxonomic groups. Here, its role in the early developmental stages of the fungal necrotroph *Botrytis cinerea* was investigated. Pharmacological analysis demonstrated that NO modulated germination, germ tube elongation and nuclear division rate. Experimental evidence indicates that exogenous NO exerts an immediate but transitory negative effect, slowing down germination-associated processes, and that this effect is largely dependent on the flavohemoglobin BCFHG1. The fungus exhibited a “biphasic response” to NO, being more sensitive to low and high concentrations than to intermediate levels of the NO donor. Global gene expression analysis in the wild-type and Δ*Bcfhg*1 strains indicated a situation of strong nitrosative and oxidative stress determined by exogenous NO, which was much more intense in the mutant strain, that the cells tried to alleviate by upregulating several defense mechanisms, including the simultaneous upregulation of the genes encoding the flavohemoglobin BCFHG1, a nitronate monooxygenase (NMO) and a cyanide hydratase. Genetic evidence suggests the coordinated expression of *Bcfhg*1 and the NMO coding gene, both adjacent and divergently arranged, in response to NO. Nitrate assimilation genes were upregulated upon exposure to NO, and BCFHG1 appeared to be the main enzymatic system involved in the generation of the signal triggering their induction. Comparative expression analysis also showed the influence of NO on other cellular processes, such as mitochondrial respiration or primary and secondary metabolism, whose response could have been mediated by NmrA-like domain proteins.

## 1. Introduction

Nitric oxide (NO) is a highly reactive diatomic molecule with a half-life of a few seconds and a major regulatory role in numerous physiological processes in organisms from different taxonomic groups [1,2,3,4,5]. Since the discovery of its synthesis by human cells at the beginning of the 20th century [6] and its role as an endogenous regulator of the vascular system in the 1980s [7], intense research efforts have been made to learn more about the metabolism and signaling pathways of this molecule in different living beings. Thus, it is known that NO is produced mainly from L-arginine by nitric oxide synthase (NOS) in animal cells and its mode of action depends on its concentration. At low concentrations, cyclic GMP (cGMP)-dependent signaling occurs via the activation of soluble guanylate cyclase, while at high concentrations, it carries out cGMP-independent signaling via mainly reversible post-translational protein S-nitrosation [8]. The molecule is thus involved in the signal transduction pathways of the nervous, immune and vascular systems [9]. In plants, where NO participates in various physiological processes such as root growth, stomatal movement, iron homeostasis, flowering or responses to biotic and abiotic stresses, enzymatic L-arginine-dependent NO synthesis has also been detected, but the major route is via the reduction of nitrite by nitrate reductase (NR) [8,9]. In fungi, NO metabolism has been less frequently explored [3]. The two previous pathways of NO synthesis have been identified in this kingdom, but the NOS-dependent oxidative pathway has been poorly characterized, and it is unclear whether sequences homologous to the canonical mammalian nitric oxide synthases are present in fungal genomes [10,11]. The reductive pathway by NR and the enzyme nitrite reductase is considered the most important enzymatic source of NO in fungi [9,12].

NO also plays a broad signaling role in fungi, participating in processes ranging from host infection and colonization to the modulation of secondary metabolism and sexual and asexual development. The molecule seems to have a versatile function in this last area, causing the inhibition or stimulation of conidiation, germination and growth, depending on the species. In *Colletotrichum coccodes*, the treatment of their spores with sodium nitroprusside (SNP), a NO donor, significantly inhibited their germination and development. This effect was counteracted by the NOS inhibitors L-NNA and L-NMMA, which even accelerated germination and development [13]. SNP suppressed the germinability of *Penicillium expansum* spores and significantly reduced their germ tube length, which was accompanied by a remarkable burst of intracellular ROS and increased carbonylated proteins [14]. This NO donor also had a fungicidal effect on the in vitro conidial germination of *Fusarium oxysporum* f. sp. *fragariae*, *F. oxysporum* f. sp. *quitoense* and *F. sulphureum*, but while the donor suppressed mycelial growth in the latter, in *F. oxysporum* f. sp. *fragariae* it was enhanced in a dose-dependent manner [15,16,17]. Another NO donor, sodium nitrite, reduced the amount of germinable spores of *Aspergillus fumigatus* at a concentration of 2 mmol/L at acidic pH, while at 5 mmol/L, the donor killed all spores. Various S-nitrosothiols, such as SNC (S-nitrosocysteine), SNAP (S-nitroso-N-acetylpenicillamine) and SNAC (S-nitroso-N-acetylcysteine), at 5 mmol/L and pH 6.5, also had a fungicidal effect, but this was less intense, and GSNO, on the other hand, stimulated spore germination [18]. Concentrations of 50–500 μL/L of gaseous NO produced an antifungal effect on the spore germination, sporulation and mycelial growth of the postharvest pathogens *Aspergillus niger*, *Monilinia fructicola* and *Penicillium italicum* in vitro, and the growth of the pathogen *Trichophyton rubrum* was inhibited in vitro by treatment with 420 nm intense pulsed light, which was linked to the upregulation of NOS and the production of excessive NO [19,20].

In contrast, the NO scavenger PTIO caused a delay in the germination of spores of the hemibiotrophic ascomycete *Magnaporthe oryzae,* and the NO scavenger cPTIO delayed germination and decreased the length of the germ tube of the urediniospores of *Puccinia striiformis Westend* f. sp. *tritici*, although a similar phenotype was produced after treatment with L-arginine [11,21]. It has also been reported that NO is highly accumulated at an early stage of in vitro conidiation of the mycoparasite *Coniothyrium minitans*, and the inhibitor of mammalian NOS enzyme L-NAME suppressed its conidiation, thus indirectly suggesting the presence of mammalian NOS-like enzymes in the fungus [22]. The percentage of germination and the relative rate of elongation of the germ tubes of *Candida albicans* blastoconidia were enhanced by treatment with SNP and SNAP. However, NO also decreased the viability of the blastoconidia. These effects were more noticeable in acidic conditions than in alkaline conditions in vitro [23]. Increased mycelial growth was also found in *Neurospora crassa* within a certain range of SNP concentrations (0.0001–0.01 mM), but increasing the SNP dose to 0.1 mM was ineffective [24]. A positive regulatory effect of NO on the growth of the fungus *Phycomyces blakesleeanus* has also been observed [25].

The same versatile regulatory effect of the molecule has been observed in pollen grain development. Thus, in *Lilium longiflorum* and *Arabidopsis thaliana*, a NO donor negatively regulated the growth rate of the pollen tube, and this response was abrogated in the presence of cPTIO [26,27]. In *Camellia sinensis*, the NO donor DEA NONOate inhibited pollen germination and pollen tube growth in a dose-dependent manner, while cPTIO and L-NNA partially reversed these effects [28]. However, a NO donor stimulated *Pinus bungeana* pollen tube growth in a dose-dependent manner [29].

Fungi also have numerous NO catabolic mechanisms that aim to combat its toxicity when the molecule reaches high concentrations that result in nitrosative and/or oxidative stress, which negatively affects physiological processes. Among them, the most ubiquitous inducible mechanism for its detoxification is that constituted by flavohemoglobins (FHbs) [9]. These are chimeric globins formed by an N-terminal globin domain and an adjacent C-terminal redox-active protein domain, and their function is described as NO dioxygenase, which requires NADPH, FAD and O_2_ to turn NO into NO_3_^−^ [30,31]. The heme group, which reacts with O_2_, is in the globin domain, while the C-terminal redox-active domain contains potential binding sites for FADH and for NAD(P)H, which participate in the recycling of the oxidized heme [32]. Instead, under anaerobic conditions, FHbs consume NO to produce N_2_O with the involvement of NADH [33,34].

In this context, the present work deals with the study of the influence of NO on the development of *Botrytis cinerea*, a necrotrophic plant pathogenic fungus that causes the disease known as gray mold and produces great losses of horticultural production worldwide. The fungus produces NO, in a regulated way, both during saprophytic growth and in planta. It has a single gene, *Bcfhg*1, that encodes a functional flavohemoglobin which constitutes its main NO detoxification system. Functional analysis demonstrated that the flavohemoglobin BCFHG1 is not a virulence factor, and it has been suggested that it could be more related to the modulation of endogenous NO levels produced by the fungus during specific developmental stages than to the protection of the fungus during the host plant infection [35,36]. Through an approach that combines the use of microscopy techniques, drugs that modulate NO levels and the development of a global expression analysis, the aim herein is to identify the physiological processes in which this gas participates in the fungus and to characterize the underlying genetic factors.

## 2. Materials and Methods

### 2.1. Fungal Strains and Culture Conditions

The *B. cinerea* strains used in this study were the wild-type strain B05.10 [37] and the mutant strain Δ*Bcfhg*1, which were obtained using a gene replacement strategy and with a lack of the NO detoxifying activity of the *Bcfhg*1 gene product [35]. Both strains were routinely grown on potato dextrose agar (PDA; Difco, Omagh, UK) plates containing 25% *w*/*v* tomato leaf extract [38]. Freshly sporulating 15-day-old cultures were used to obtain conidia in all experiments.

To prepare the conidia suspension, spores were collected from the plate by moistening the mycelium with water and gently rubbing its surface with a glass spreader. Subsequently, the conidia suspension was filtered through glass wool, and the filtrate was washed three times with water. In all the tests carried out in this work, the PDB medium was used at half the concentration established by the manufacturer, PDB ½ (Potato Dextrose Broth, Difco, UK), adjusting the concentration of the spores to 5 × 10^5^ spores/mL.

### 2.2. Germination Assays

To analyze the effect of NO on the germination and elongation of the germinative tube of fungal spores, 60 μL drops of the suspension in PDB ½ at a concentration of 5 × 10^5^ spores/mL were used together with different chemical modulators of NO levels: diethylenetriamine (DETA NONOate, Sigma Aldrich, Steinheim, Germany) was used as the NO donor at a final concentration of 250 µM (or as indicated), Nω-nitro-L-arginine (L-NNA, Sigma Aldrich, Steinheim, Germany) as an inhibitor of nitric oxide synthase (NOS) activity at a final concentration of 1 mM and 2-(4-carboxyphenyl)-4,4,5,5-tetramethylimidazoline-1-oxyl-3-oxide (cPTIO, Sigma Aldrich, Steinheim, Germany) as a nitric oxide scavenger at a final concentration of 500 µM. Two types of tests were differentiated depending on the moment of the addition of the corresponding chemical agent: in the first one, the chemical agents were in contact with the spores from the beginning of the cultures, and in the second one, the chemical agents were added to the cultures after a period of 4 h of incubation, which was necessary for the spore germination program to be launched. In both cases, the drops were deposited in Petri dishes that were placed in closed plastic boxes with the bottom covered with moistened paper to prevent the evaporation of the drops. The boxes were kept at 22 °C in the dark and without agitation until imaging with an optical microscope. At the times indicated according to the experiment (2, 3, 4, 6, 8 and 10 h), the germination percentage and the length of the germ tube were quantified, and the spores were classified into three developmental stages according to the length of their germ tubes: spores in which the primordium of the germination tube had not been emitted (stage 0); spores in which the germination tube had been emitted, but the length was smaller than the diameter of the ungerminated spore (stage 1); and spores whose germ tube had a length greater than the diameter of the ungerminated spore (stage 2).

### 2.3. Nuclei Staining

To determine the influence of NO on the fungal nuclear division, 15 mL suspension in PDB ½ at a concentration of 5 × 10^5^ spores/mL was placed in Petri dishes and incubated at 22 °C in the dark and without agitation until staining (0, 4, 6 and 8 h). For this, the cultures were collected by centrifugation. The pellets were then washed twice with 1 mL Milli-Q water, and 1 mL of the fixing solution (100% ethanol and acetic acid in a 3:1 ratio (*v*/*v*)) was added to each tube. The samples were thus preserved at 4 °C overnight and washed twice with 1 mL Milli-Q water. The spores were resuspended in 1 mL Milli-Q water, to which two drops of the commercial preparation of the DAPI dye agent (4′,6-diamino-2-phenylindole) (NucBlue^®^ Fixed Cell Stain ReadyProbes^TM^ reagent, Life Technologies^TM^, Grand Island, NY, USA) were added, leaving it to act for 10 min at room temperature. The samples were washed twice with Milli-Q water and the spores were finally resuspended in 1 mL 25% glycerol. For observation under a fluorescence microscope, 10-microliter aliquots were used.

### 2.4. Microscopy

For the visualization of the fungal structures, an optical microscope (LEICA DM LB optical microscope, Leica Microsystems, Wetzlar, Germany) was used, applying a red suppression filter (BG38, blue filter) under the illumination of an EBQ 100 isolated-L/131-26B light source (Leistungselektronik JENA GmbH, Jena, Germany) with a mercury vapor bulb (Osram, Munich, Germany) when the excitation of a fluorescent dye and the consequent visualization of the emitted fluorescence were necessary. The images were captured using a LEICA DC300F digital camera (Leica Microsystems) with the help of the Leica IM 1000 Image Manager software (Leica Microsystems). ImageJ 1.47V and Fiji software (National Institutes of Health, Bethesda, MD, USA) were used to process the images and perform the different counts and measurements of the fungal structures [39,40].

### 2.5. Statistical Analysis

The data generated in the germination assays were analyzed by applying a Student’s *t* analysis at *p* < 0.05 or a one-way analysis of variance (ANOVA) followed by a Bonferroni test at *p* < 0.05, performed on Statgraphics Centurion XVI software (Statgraphics Technologies, Inc., Plains, Virginia).

### 2.6. Differential Expression Analysis

To analyze the effect of NO on the transcriptome of the germinating spores, 25 mL suspension in PDB ½ medium at a concentration of 5 × 10^5^ spores/mL was inoculated in square plates of 120 mm side size. Both strains, the wild-type BO5.10 and the Δ*Bcfhg*1 mutant, were handled and treated in identical conditions and samples were collected in parallel. The cultures were incubated at 22 °C in the dark and without agitation for 4 h to allow the activation of their germination program. After this time, the spores of a fraction of the cultures were collected in a 50-milliliter tube, where the donor DETA was added at a final concentration of 250 µM. The mixture was returned to the plates and the incubation was extended for two more hours before harvesting. The spores of the other fraction were kept in continuous incubation over the total test period of 6 h as a control. To collect the spores, the bottom of the square Petri dish was scraped with a glass spreader, and the content was transferred to a 50-milliliter tube. After pelleting the samples by centrifugation and washing them twice in Milli-Q water, they were resuspended in 500 µL of RNAlater^®^ solution buffer (Invitrogen, Waltham, MA, USA) and incubated for 1 h at 4 °C. Finally, the mixture was centrifuged, and the pellet was frozen in liquid nitrogen and stored at −80 °C. Three independent biological experiments were carried out in triplicate for each of the samples.

The tissue samples were submitted to the North Carolina State Genomic Sciences Laboratory (Raleigh, NC, USA) for RNA extraction, Illumina RNA library construction and sequencing. Total RNA was extracted from tissues using the Qiagen RNeasy Plant Mini Kit (Qiagen, Germantown, MD, USA) according to the manufacturer-supplied protocol. Prior to library construction, RNA integrity, purity and concentration were assessed using an Agilent 2100 Bioanalyzer with an RNA 6000 Nano Chip (Agilent Technologies, Santa Clara, CA, USA). The purification of messenger RNA (mRNA) was performed using the oligo-dT beads provided in the NEBNext Poly(A) mRNA Magnetic Isolation Module (New England Biolabs, Ipswich, MA, USA). Complementary DNA (cDNA) libraries for Illumina sequencing were constructed using the NEBNext Ultra Directional RNA Library Prep Kit (NEB) and NEBNext Mulitplex Oligos for Illumina (NEB) using the manufacturer-specified protocol. Briefly, the mRNA was chemically fragmented and primed with random oligos for the first-strand cDNA synthesis. Second-strand cDNA synthesis was then carried out with dUTPs to preserve strand orientation information. The double-stranded cDNA was then purified, end repaired and “a-tailed” for adaptor ligation. Following ligation, the samples were selected at a final library size (adapters included) of 400–550 bp using sequential AMPure XP bead isolation (Beckman Coulter, Brea, CA, USA). Library enrichment was performed, and specific indexes for each sample were added during the protocol-specified PCR amplification. The amplified library fragments were purified and checked for quality and final concentration using an Agilent 2200 Tapestation with a High Sensitivity DNA chip (Agilent Technologies, USA). The final quantified libraries were pooled in equimolar amounts for clustering and sequencing on one flow cell lane of an Illumina HiSeq 2500 DNA sequencer utilizing a 125-bp single-end sequencing reagent kit (Illumina, San Diego, CA, USA). The software package Real Time Analysis (RTA) was used to generate raw bcl, or base call, files, which were then de-multiplexed by sample into fastq files.

Fastq read quality was assessed using the FastQC High Throughput Sequence QC Report (v.0.11.7). The reads were then mapped onto the *B. cinerea* genome (B05.10 ASM14353v4) using Hisat2 (v.2.1.0). For each library, the number of reads mapped onto each gene was counted using StringTie -eB (v.1.3.5). Differential expression analysis between the pairwise comparisons of interest was performed with the DESeq2 (v.1.28.1) R package, considering the genes as differentially expressed (DEG) when log_2_foldchange > 2 and *p*_adjusted_ ≤ 0.05. Gene ontology (GO) enrichment analysis was performed on the sets of DEGs of two of the three comparisons analyzed (non-exposed B05.10/NO-exposed B05.10 and non-exposed Δ*Bcfgh*1/NO-exposed Δ*Bcfgh*1). The matrix containing the total set of genes from *B. cinerea* was annotated using InterProScan Version 5.52-86.0 [41]. Fisher’s exact tests were performed on the GO terms associated with the significantly up- and downregulated genes (adjusted *p*-value of < 0.05, any fold change), and REVIGO was used to trim the resulting lists of significantly overrepresented GO terms [42]. The DEGs obtained in the three comparisons were analyzed via manual inspection using the browsers NCBI (https://www.ncbi.nlm.nih.gov/ (accessed on 10 June 2021)), EnsemblFungi (http://fungi.ensembl.org/index.html (accessed on 15 June 2021)), *Botrytis cinerea* Portal (https://bionfo.bioger.inrae.fr/botportalpublic (accessed on 15 June 2021)) and Joint Genome Institute (https://genome.jgi.doe.gov/portal/ (accessed on 20 June 2021); PMID 24225321); the databases InterProScan Version 5.52-86.0 (https://www.ebi.ac.uk/interpro/search/sequence/ (accessed on 20 June 2021)) and Uniprot (Universal protein) [43]; the automatic annotation server BlastKOALA (https://www.kegg.jp/blastkoala/ (accessed on 25 June 2021)) and the software Blast2go [44].

## 3. Results

### 3.1. Pharmacological Analysis

#### 3.1.1. Modulating NO Levels Affects Germination of *B. cinerea*

A preliminary assessment of the influence of NO on the germination of the *B. cinerea* B05.10 strain was performed by exposing the macroconidia suspensions to the NO-modulating drugs DETA NONOate, a NO producer; cPTIO, a NO scavenger; and L-NNA, an inhibitor of mammalian NOS enzyme. The effect of the chemicals on the germination of macroconidia of the flavohemoglobin-deficient mutant strain Δ*Bcfhg*1 was analyzed in parallel. Estimations of germination were carried out after 3 h (Figure 1), the moment within the temporal window when most spores of both strains germinate, and at 10 h a time point at which all the spores have already germinated. The germination rate of the wild-type strain was reduced by exposure to DETA, as the percentage of spores in stage 0 increased and the percentages of spores in stage 1 and stage 2 decreased in comparison with the situation observed in the absence of the NO donor (Figure 1a). This behavior was restored when 250 μM DETA and 500 μM of cPTIO were added simultaneously. On the other hand, when B05.10 spores were incubated in the presence of 500 μM of cPTIO, a slight increase in the percentage of spores in stage 2 in comparison with the reference situation in the absence of DETA was observed. The addition of 1 mM of L-NNA appeared to slightly accelerate germination.

No significant differences were observed between the Δ*Bcfhg*1 strain and the wild-type strain in their behavior in the absence of DETA. However, net differences were observed between both strains in the presence of NO, as the addition of 250 μM of DETA caused a total blockage of the germination process of the mutant strain after 3 h (Figure 1b). The simultaneous addition of DETA and cPTIO partially reverted this effect. The addition of cPTIO did not determine any change, while the addition of L-NNA also slightly accelerated germination in the mutant strain. 

All the B05.10 spores reached stage 2 after 10 h in all conditions tested (results not shown), suggesting that NO causes a delay in germination but does not determine a loss of viability of the spores. The spores of the mutant strain could not overcome the effect of NO (only 3% of the spores had germinated after 10 h) (results not shown), indicating that BCFHG1 is essential to detoxify the exogenous NO and that DETA maintains effective concentrations of NO in the culture medium for at least the first 10 h of incubation.

#### 3.1.2. Quantification of the Effect of NO along Time

Once it was detected that NO negatively influences germination, its effect was quantified on both strains as a function of time. We first characterized the germination kinetics of the reference strain, B05.10, when cultured in the experimental conditions considered in our analysis (PDB ½ in static culture). Although germination was not fully synchronous in these conditions, most spores germinated between 2 (8%) and 4 h (83%). At the later time point, spores in stage 1 predominated (52%) and spores in stage 2 represented 31%. After 6 h, 98% of the spores had germinated, most of them (90%) having reached stage 2 (Figure 2a). The germination kinetics of strain Δ*Bcfhg*1 were the same as those observed for the wild-type reference strain, and no significant difference was detected between the samples in equivalent stages at the corresponding time points (Figure 2c).

To quantify the effect of NO on fungus development, the NO donor was added to a fraction of the cultures when they were established, and the germination of both strains was evaluated at the time points considered to characterize their germination kinetics. In the case of the wild-type spores (Figure 2a), the reduction in the percentage of spores in stage 1 after 2 h in the presence of DETA indicated that the initiation of germination was delayed. This delay was obvious after 4 h when, the percentage of spores in stage 1 being similar to that observed for this time point in the absence of DETA, the percentage of spores in stage 0 was much higher and the percentages of spores in stage 2 was much lower. After 6 h, spores in stage 2 already predominated, and after 8 h, most of the spores had reached this stage, as in the absence of DETA. In the case of the Δ*Bcfhg*1 strain (Figure 2c), cultivation in the presence of DETA completely blocked germination, and this blockage lasted for at least a 10-h period, the time considered in this analysis.

#### 3.1.3. The NO Influence Depends on Its Concentration

As the two strains analyzed responded differently to the NO donor at the concentration used in the previous analysis, we next evaluated their sensitivity threshold by permanent exposure to increasing concentrations of the NO donor. The quantification of germination percentages at 4 h and of the germ tube length at 10 h showed that the fungus responds differently to different concentrations of NO, exhibiting a “biphasic response” to the molecule. This was characterized by a first response in which a strong repressive effect on the germination and elongation of the germ tube was determined at low DETA concentrations (7.5 μM), which was notably attenuated at intermediate concentrations (15–30 μM). The second response appeared once this concentration was exceeded and caused a dose-proportional inhibition up to the highest concentration tested. The mutant strain was drastically affected in its germination even at low concentrations of DETA, being unable to germinate when the donor concentration was greater than 15 μM (Figure 3a,b).

#### 3.1.4. Effect of NO on Nuclear Division during Germination of *B. cinerea* B05.10 and Δ*Bcfhg*1 Strains

The alterations in the germination process observed in both strains by NO suggest a possible affectation of nuclear division. To address this question, the number of nuclei was estimated in spores during the early stages of germination (4 h incubation in the experimental conditions used before) after permanent exposure or not to the NO donor. In both strains, the average number of nuclei at 0 h was four. This value almost doubled (seven) after 4 h of incubation in the absence of NO. In the presence of DETA, the average number of nuclei in the B05.10 strain at 4 h was five, indicating that exogenous NO reduces the nuclear division rate. In the Δ*Bcfhg*1 strain, after 4 h of incubation in the presence of DETA, the mean number of nuclei was the same as that counted at 0 h. Therefore, exogenous NO appears to paralyze nuclear division in the flavohemoglobin-deficient strain (Figure 4).

#### 3.1.5. Effect of Exposure to NO Once the Germination Program Has Been Launched

The effect of NO in the initial stages of development in *B. cinerea* was analyzed when the spores were exposed permanently to NO. We then set out to find out the degree of immediacy with which NO exerts its effect. For this purpose, the spore suspensions of the B05.10 and Δ*Bcfhg*1 strains were incubated for 4 h in the experimental conditions used before. At this time point, when the germination program had been launched and most spores had already germinated, the NO donor was added, and the incubation was extended (Figure 5a and Figure 6a). Samples for evaluation were taken 2 and 4 h later (time points 6 and 8 h after the cultures were established). 

As shown in Figure 5b,c, the addition of DETA slowed down the germination of the B05.10 strain spores. This effect appeared to be transitory, since after 8 h, essentially all the spores were also stage 2 in the NO-exposed samples. The quantification of the germ tube length indicated that the germ tube growth rate was not constant, with elongation occurring more slowly in the first time interval considered, between 4 and 6 h, and faster during the second interval, between 6 and 8 h. The addition of DETA drastically reduced the elongation rate during the first interval after the addition of DETA, but during the second interval, growth resumed and the elongation rate reached almost normal values (Figure 5d,e). Nuclei number counting indicated that the nuclear division rate was not constant either during germination or in the early stages of germ tube elongation, being faster between 6 and 8 h than between 4 and 6 h (Figure 5f,g). Immediately after the addition of DETA, the nuclear division rate was also drastically reduced and, paralleling the situation observed regarding the germ tube elongation rate quantification, the nuclear division rate recovered to almost normal values in the second interval considered. Taken together, these results indicate that the exogenous NO provided by DETA to the B05.10 spores, once its germination program has been launched, exerts an immediate, temporary, short-term germination-delaying effect in *B. cinerea*.

As shown previously, the germination kinetics of the Δ*Bcfhg*1 strain were very similar to those of the B05.10 strain in the absence of DETA (Figure 5b and Figure 6b). The addition of DETA blocked germination completely, and the germination program appeared to be frozen at the phase that the spores had reached before the NO donor was added. This effect lasted for at least the first 4 h after the addition of DETA (Figure 6c). The germ tube elongation pattern of the Δ*Bcfhg*1 strain in the absence of DETA was the same as that of B05.10 (no significant differences were detected between the corresponding intervals in the two strains). The addition of DETA reduced the growth rate to negligible values during the first interval. In contrast to the B05.10 strain, this inhibitory effect on germ tube growth was maintained during the second interval (Figure 6d,e). The nuclear division kinetics of the Δ*Bcfhg*1 strain in the absence of DETA were also the same as those of the B05.10 strain, but the addition of DETA froze nuclear division (Figure 6f,g). Therefore, strain Δ*Bcfhg*1 behaved in the same way as the wild-type B05.10 strain during germination and the first stages of germ tube elongation in the absence of DETA, but exogenous NO froze the germination program in the flavohemoglobin-deficient strain. 

### 3.2. Differential Gene Expression Analysis in Response to NO Exposure

A comparative transcriptomic analysis in response to exogenous NO both in the B05.10 wild-type strain and in the Δ*Bcfgh*1 mutant strain was carried out in the experimental conditions described above. Thus, mycelium samples for RNAseq were collected for both strains from 6 h non-exposed cultures and from cultures exposed to NO after 4 h. RNAseq analysis was performed on the four samples, each in three biological replicates, except for the sample of NO-exposed mutant spores, in which one replicate was discarded for failing the quality test. The total number of reads for each library and the mapping efficiencies are presented in Appendix A. Two comparisons between samples were then considered: (1) non-exposed B05.10/NO-exposed B05.10 in order to characterize the response of the wild-type strain to NO, and (2) non-exposed Δ*Bcfhg*1/NO-exposed Δ*Bcfhg*1, which would allow the identification of factors specifically responding to NO in the system deprotected against nitrosative stress. To discriminate between genes responding to NO in the mutant strain and genes whose expression is altered due to the mutation itself, the expression profiles of the B05.10 and the Δ*Bcfhg*1 strains non-exposed to NO were compared. Figure 7 summarizes the number of differentially expressed genes (DEGs) in each comparison. In total, 54 genes were downregulated and 88 were upregulated in the B05.10 strain due to its exposure to NO. Only two genes were downregulated and two were upregulated when comparing the B05.10 and Δ*Bcfhg*1 expression profiles at the 6-h developmental stage. The response to NO in the Δ*Bcfhg*1 strain was stronger than that in the wild-type strain, the total number of DEGs being 422–157 downregulated and 265 upregulated. The complete lists of DEGs in each comparison are presented in Appendix A. 

#### 3.2.1. Differential Gene Expression Analysis of the B05.10 Strain Response to NO

Gene ontology (GO) enrichment analysis performed on the upregulated DEGs in the comparison involving the B05.10 wild-type strain showed the overrepresentation of 19 terms, including the oxidation-reduction process, nitrate metabolic process, reactive nitrogen species metabolic process, nitric oxide biosynthetic process, nitric oxide metabolic process, reactive oxygen species biosynthetic process, siroheme biosynthetic process, sulfate assimilation and DNA catabolic process. Enriched downregulated processes included terms related to ion homeostasis, cation homeostasis, inorganic ion homeostasis, the oxidation-reduction process, the glutamate metabolic process, chemical homeostasis and the isoprenoid biosynthetic process (Appendix A).

The manual inspection of the lists of DEGs showed that the largest group among the upregulated genes consisted of those associated with the GO terms of the biological process and molecular function of the “redox process” and “oxidoreductase activity”, respectively, including catalase *Bccat*6 (Bcin05g04580). It is worth noting that four genes related to the term polyketide synthase (Bcin05g08390, Bcin14g01070, Bcin10g01470 and Bcin01g11530) and one gene related to the term “polyketide cyclase” (Bcin14g01350) were identified. Several genes encoding proteins related to transmembrane transport were also induced, as well as the chitin deacetylase coding gene *Bccda*1 (Bcin03g05710). Only one gene, Bcin05g04650, the ortholog of the *A. nidulans abaA* gene, annotated as a transcription enhancer factor with the TEA/ATTS DNA-binding domain (PF01285) was identified related to gene regulation activity.

Several nitrogen metabolism genes were found. *BcniiaA* (Bcin01g05790) encoding nitrite reductase and *BcniaD* (Bcin07g01270) encoding nitrate reductase were upregulated, indicating that the molecule needed to signal their expression, NO_3_^−^, was generated. The flavohemoglobin coding gene *Bcfhg*1 (Bcin04g06230) and its adjacent gene Bcin04g06240, encoding a nitronate monooxygenase (NMO), also appeared to be upregulated. Both enzymes might contribute to the generation of NO_3_^−^ derived from their activities on NO or on nitroalkanes, respectively. In addition, the genes encoding a cyanide hydratase (Bcin12g06180), a nitrilase (Bcin13g05670), two amidases (Bcin03g08640 and Bcin03g05590) and three NmrA-like domain-containing proteins (Bcin11g06310, Bcin08g04910 and Bcin03g00700) were found. One of these (Bcin11g06310) presented the highest induction level in this comparison (9.9). However, the regulatory genes *BcareA* (Bcin09g04960) and *Bcnit*4 (Bcin12g04500) and the nitrate transporter *BccrnA* (Bcin01g06290) were not detected as DEGs (Figure 8A).

Among the downregulated genes, the succinate dehydrogenase/fumarate reductase flavoprotein (Bcin05g04870) that participates in the TCA cycle and the glutamate decarboxylase *Bcgad*1 (Bcin09g05020), which is involved in the GABA shunt [45], were identified. A glutamine amidotransferase (Bcin07g02370) that participates in nitrogen metabolism was the only observed representative of this category. Genes encoding a cfem domain-containing protein (Bcin16g02840, related to pathogenicity, sporulation and stress response); the necrosis-inducing protein BcNEP2 (Bcin02g07770); the GAS2 protein (Bcin14g04260), which is regulated by *Bmp*1 MAP kinase cascade, a hydrolase of cutinase type (Bcin13g05760); the aspartic proteinase *Bcap*8 (Bcin12g02040); and the *Bccpd*3 gene (Bcin01g01590), a cyclophilin-dependent gene related to development and virulence, were also found. A summary of the DEGs and processes affected by NO in the wild-type strain referred to in this section is presented in Appendix A.

#### 3.2.2. Differential Gene Expression Analysis of the Δ*Bcfhg*1 Strain Response to NO

Both in terms of the number of up- or downregulated genes as well as the levels of induction/repression determined, the response was stronger in the Δ*Bcfhg*1 strain than in the wild-type strain (Figure 7, Appendix A). To evaluate the potential effect that the deletion of the *Bcfhg*1 gene might have on the transcriptomic profile of the germinating fungal spores, the comparison between non-exposed B05.10 and non-exposed Δ*Bcfhg*1 after 6 h was considered first. Four DEGs were detected (Figure 7, Appendix A). The flavohemoglobin coding gene *Bcfhg*1 (Bcin04g06230) was downregulated, as expected. Surprisingly, the genes Bcin04g06240 and Bcin04g06250, closely linked to *Bcfhg*1 and divergently arranged in relation to it, appeared to be upregulated. A second downregulated gene was detected, Bcin10g05210, related to the GO term “integral component of membrane” (Figure 8C).

Excluding these genes, a GO enrichment analysis of DEGs in the comparison of non- exposed Δ*Bcfhg*1 and NO-exposed Δ*Bcfhg*1 was carried out. Among the upregulated genes, enrichment was detected in terms such as the oxidation-reduction process, cellulose metabolic process, beta-glucan catabolic process and polysaccharide metabolic process, from a total of 11 enriched GO terms. The 23 enriched downregulated processes included terms related to the glycogen metabolic process, cellular glucan metabolic process, cellular carbohydrate metabolic process, oxidation-reduction process, carbohydrate metabolic process, biotin metabolic process, polysaccharide metabolic process, glycerol metabolic process, phosphate ion transport and alditol metabolic process (Appendix A). The manual inspection of the lists of genes allowed the identification and/or categorization of a total of 224 DEGs (153 upregulated and 71 downregulated). Of the 265 genes that presented an induction response in this comparison, 59 were common to the comparison of the non-exposed B05.10 and NO-exposed B05.10 (Appendix A), while the remaining 206 were exclusive to the comparison of the non-exposed Δ*Bcfhg*1 and NO-exposed Δ*Bcfhg*1 (Appendix A). Appendix A shows the genes exclusive in the comparison of the non-exposed and NO-exposed B05.10 strain. 

Among the nitrogen metabolism genes, those encoding the cyanide hydratase (Bcin12g06180), the amidase enzyme (Bcin03g08640) and the three NmrA-like domain-containing proteins induced in the wild-type strain were also detected in the mutant strain, in most cases displaying now higher levels of induction (Figure 8B). The nitrilase Bcin13g05670 and the amidase Bcin03g05590, induced in the wild-type strain, were not differentially expressed in the mutant strain, but another nitrilase, Bcin05g04960, and another amidase, Bcin12g00040, were specifically induced in Δ*Bcfhg*1. It is interesting to note that the *Bcnii*A (Bcin01g05790) and *Bcnia*D (Bcin07g01270) genes were not found among the DEGs in response to NO in the Δ*Bcfhg*1 strain, indicating that their induction in response to NO is dependent on the activity of the flavohemoglobin BCFHG1. Bcin04g06240, the gene adjacent to *Bcfhg1* and encoding an NMO, did not respond to NO either (see Figure 8B). Finally, a deaminated glutathione amidase (Bcin11g00510) and many additional NmrA-like domain-containing proteins (Bcin09g04560, Bcin02g01880, Bcin01g03270, Bcin05g05290, Bcin09g00910, Bcin02g05800 and Bcin12g00030) were found.

Most genes related to redox activity which were differentially induced in the wild- type strain were also induced in the mutant strain. Additional genes within this category appeared to be specifically induced in the mutant strain, including *Bcnde*4 (Bcin03g08570) encoding the external NADH dehydrogenase 4 and the cytochrome c oxidase copper chaperone *Bccox*17 (Bcin02g03350), indicating alterations in the electron transport chain in the mitochondria. Remarkably, a putative glutamate-cysteine ligase regulatory subunit coding gene (Bcin14g01940), related to the first step in the cellular GSH biosynthesis pathway, and several glutathione S-transferases, *Bcgst*1, *Bcgst*3, *Bcgst*5, *Bcgst*13 and *Bcgst*14, were induced. The cyclophilin-dependent gene *Bccpd*1 (Bcin07g07020), related to morphogenesis and virulence, and the chitin deacetylase gene *Bccda*1 (Bcin03g05710), which was also induced in the wild-type strain, were upregulated. 

Genes related to gene regulation processes were more abundant in the list of genes induced in the mutant strain, which included several genes encoding transcription factors of different types: Bcin01g10220, encoding a serine threonine kinase; Bcin02g03440, encoding a chromo domain-containing protein associated with the alteration of the structure of chromatin to the condensed morphology of heterochromatin; and Bcin02g04870, encoding a histone involved in nucleosome assembly. Secondary metabolism genes participating in different pathways were also more abundant among the induced genes.

Among the repressed genes, 36 were common to the two comparisons (Appendix A). In total, 121 genes were repressed specifically in the mutant strain (Appendix A). A number of primary metabolism genes were found in this group, including a glutamate decarboxylase coding gene (Bcin03g01040). It is interesting to note that *Bcgph*1 (Bcin15g03620), encoding a glycogen phosphorylase, and *Bcgdb*1 (Bcin01g10310), encoding a glycogen debranching enzyme, were identified within the latter group, indicating a downregulation of glycogen metabolism and mobilization. Similarly, Bcin09g04240, encoding glycerol kinase, and Bcin11g04710, encoding glycerol 2-dehydrogenase, appeared to be repressed, suggesting that glycerol metabolism is also downregulated in this condition. Two transcription factors were repressed, and remarkably, numerous genes were involved in different secondary metabolism pathways, such as the thioesterase coding gene *Bcboa*10 (Bcin01g00100), involved in the synthesis of botcinic acid; the tetrahydroxynaphthalene reductase *Bcbrn*1 (Bcin04g04800), involved in the synthesis of melanin [46]; and the polyketide synthase gene *Bcpks*7 (Bcin10g00040). A summary of the DEGs and processes affected by NO in the Δ*Bcfhg*1 strain referred to in this section is presented in Appendix A.

## 4. Discussion

The signaling function of NO is widely extended in the cells of different living beings, including animals, plants and microorganisms [2,47,48,49,50,51]. Our work has focused on the analysis of the influence that the molecule could have on development in *B. cinerea* and has shown that the variation in NO levels affects the dynamics of macroconidia germination and the associated processes of germ tube elongation and nuclear division. This effect has turned out to be largely dependent on the flavohemoglobin enzyme BCFHG1.

A variable regulatory effect of the molecule on germination in other organisms has been previously described. In some species, as in the plant genera *Lilium*, *Arabidopsis* and *Camellia* and the fungus *C. coccodes*, NO showed a behavior similar to that shown here, while in others, such as *P. bungeana*, *M. oryzae* and *P. striiformis Westend*, its effect was the opposite, stimulating the germination process of pollen grains or spores [11,13,21,26,27,28,29]. All these observations present NO as a radical with a versatile and conserved regulatory role in the early development of fungi and plants, for which the different species could have differences in the response to fluctuating levels of the molecule.

Our pharmacological analysis focused mainly on the effect of exogenous NO using the DETA donor, but it was also found that the application of L-NNA determined a slight increase in germination rates in both strains in the absence of exogenous NO. In this regard, the ability of the fungus to produce NO both in germinating conidia and mycelium in saprophytic growth and during host invasion is known [36,52,53,54]. Our results confirm this capacity and suggest a possible enzymatic origin of at least a fraction of this NO in which one or more enzymes functionally related to mammalian nitric oxide synthases would be involved, although no ortholog of genes encoding these synthases has been found in the *B. cinerea* genome. On the other hand, in previous studies by our research group, we observed that the flavohemoglobin coding gene is expressed increasingly during germination, declining shortly after during saprophytic growth, but that exposure to exogenous NO increases its expression [35]. In the present work, we also detected this inducing effect of the expression of *Bcfhg*1 by NO. Taken together, these observations lead us to consider the possibility that this endogenous NO produced when germination takes place exerts a repressive effect on this process and that the activity of the flavohemoglobin at these specific stages of development could constitute a positive regulation mechanism that allows the advancement of the germination process.

The exposure of germinating spores to NO causes an immediate but transitory effect, slowing down germ tube elongation and nuclear division rates. Such a rapid response must be mediated by a reduced number of elements that participate in a reduced number of stages. It is known that the activity of proteins involved in signal transduction can be altered under stress conditions through NO-dependent post-translational modifications leading to the nitration of Tyr residues or S-nitrosylation of Cys residues [3,55]. In this latter reaction, the resulting nitrosothiols are highly labile and dependent on the redox state of the cellular environment due to their reactivity with intracellular reducing agents. This instability translates into half-lives that range from seconds to a few minutes and, therefore, represent a very sensitive mechanism for the regulation of cellular processes [56]. In the situation of oxidative and nitrosative stress that *B. cinerea* spores must experience due to the burst of NO supplied by the donor, NO could affect, through S-nitrosylation, factors involved in the coordination of early development processes and related to cell cycle and nuclear division control and germ tube elongation. It is interesting to note that alterations in cell cycle progression have been reported by several groups upon exposure to NO-generating compounds [57,58,59]. In *S. pombe*, exposure to the NO donor DETA NONOate resulted in mitotic delay and G2/M checkpoint activation [60]. The authors found that nitrosative stress results in the inactivation of the Cdc25 phosphatase through S-nitrosylation. This causes the retention of Cdc2, the only cyclin-dependent kinase in *S. pombe* and actively required for G2/M transition, in its inactive phosphorylated form. 

Regardless of the nature of the possible targets of NO under the experimental conditions evaluated, it can be proposed that in wild-type cells, S-nitrosylation modifications can be spontaneously reversed, at least partially, and in short periods of time, particularly in the presence of an efficient NO removal system, such as BCFHG1, which maintains the redox state of the cell within limits compatible with proper cell function. In contrast, in mutant cells lacking this mechanism, the level of oxidative and nitrosative stress should be higher and more persistent. This situation would prevent the initially established S-nitrosylation-type modifications from being reversed. 

The action of this mechanism would depend on the concentration of the molecule presenting two different sensitivities and separated by a concentration of between 15 and 30 μM of the donor. It is known that the effect of NO can vary depending on its concentration, and results similar to ours have been previously observed in other fungi. Thus, sodium nitroprusside (SNP), a NO donor compound, showed an inducing effect on the formation of macroconidia and the extension of the basal hyphae, which gradually increased with between 0.001 and 0.005 and 0.001 and 0.01 mM of the donor, respectively, in cultures of *Neurospora crassa*. Exceeding these values, the treated samples presented a behavior closer to that of the control for both parameters [24]. In contrast, the same donor did not have a significant effect at concentrations ranging from 0.1 to 1 mmol/L on the spore germination of *Penicillium expansum*, but an inhibitory effect was gradually enhanced with an increasing SNP concentration up to 6 mmol/L [61]. These observations suggest a fine tuning of NO in the processes of germination and growth in fungi whereby, at low concentrations, exogenous NO would act as a signaling molecule, while at high concentrations, the induction of the formation of ROS would trigger oxidative damage that would negatively affect these processes [3].

The analysis of the transcriptomic response of the Δ*Bcfhg*1 strain to NO yielded almost three times more DEGs than that of the wild-type spores, overlapping only 22% of them. The manual inspection of the DEGs indicated that the cell redox status was affected and that a situation of oxidative stress was determined that appeared to be much more intense in the mutant strain. In addition to *Bccat*6 and a possible thioredoxin, which are already differentially upregulated in the wild-type strain (although now at a higher level), a peroxidase—the regulatory subunit of the glutamate-cysteine ligase, which is related to the first step in the cellular GSH biosynthesis pathway—and up to five GSTs (*Bcgst*1, *Bcgst*3, *Bcgst*5, *Bcgst*13 and *Bcgst*14) were strongly upregulated in the Δ*Bcfhg*1 strain. This also determined a situation of nitrosative stress that wild-type cells try to alleviate by upregulating mechanisms to directly remove NO, such as the flavohemoglobin BCFHG1, or to metabolize harmful NO-derived products. Nitroalkanes are formed from lipid nitration during nitro-oxidative stress, and NMOs catalyze their oxidative denitrification [62]. In *M. oryzae*, five NMOs were annotated, and one of them, NMO2, was shown to be required for mitigating damaging lipid nitration under nitrosative stress conditions and, concomitantly, to generate nitrate. In the *B. cinerea* genome, five NMO coding genes were found, Bcin15g05110, Bcin07g00680, Bcin05g003979, Bcin03g04860 and Bcin04g06240, but they were not functionally characterized. The latter appeared to be induced in the B05.10 spores when exposed to NO, and interestingly, its coding gene was closely linked to the *Bcfhg*1 gene. The expression of nitrilase/cyanide hydratase coding genes was also induced, suggesting that nitriles, such as cyanide, are generated upon exposure to NO. The biological degradation of cyanide/nitriles to the corresponding acid can take place via a one-step process, as exemplified by nitrilases and cyanide dihydratases (CDHs), or via a two-step process with an amide intermediate, as is the case with nitrile hydratases and cyanide hydratases (CHs) [63]. These enzymes are part of the nitrilase superfamily [64]. In *B. cinerea*, there were four genes annotated as nitrilase coding genes and one annotated as a cyanide hydratase coding gene, but none of them were functionally characterized. Bcin12g06180, the gene coding for a cyanide hydratase, was strongly induced both in the wild-type and the Δ*Bcfhg*1 strain, and interestingly, an amidase coding gene (Bcin03g08640) was also induced in both strains, suggesting that the two-step process constitutes the main cyanide/nitrile enzymatic degradation system activated in response to NO in *B. cinerea*. It is interesting to note that two of the four nitrilase coding genes appeared to respond differently, likely depending on the effective NO concentration within the cell: Bcin13g05670 was only upregulated in the wild-type and Bcin05g04960 in the mutant strain. The other two nitrilase coding genes did not respond to NO. 

*Bcfhg*1 (Bcin04g06230) and the NMO coding gene (Bcin04g06240) were induced simultaneously in response to NO. Given their functions, it can be assumed that their coordinated expression would facilitate the response against nitrosative stress. The expression of *Bcfhg*1 was abolished in the Δ*Bcfhg1* strain in the presence of DETA, as expected since it is a deletion mutant. However, surprisingly, the induction response of Bcin04g06240 was also lost in the mutant strain, which should be experiencing a more intense nitrosative stress situation. Interestingly, while *Bcfhg*1 is identified as a repressed gene in the Δ*Bcfhg*1 strain when the wild-type and mutant transcriptomes in the absence of NO are compared, Bcin04g06240 showed a high level of induction. Additionally, Bcin04g06250, encoding a putative amylase-type hydrolase and located immediately downstream, was highly induced in the mutant strain in this comparison. The expression pattern of these two genes was, therefore, altered due to the deletion of the flavohemoglobin coding gene. Bcin04g06230 and Bcin04g06240 were closely linked (separated by 769 pnt) and divergently orientated. This organization has been described in fungi facilitating the coregulated expression of genes sharing regulatory sequences in the common promoter region, as is the case of the *nia*D and *nii*A genes of *A. nidulans* or the *btp*1 and *Bcgst*II genes of *B. cinerea* [65,66]. It is tempting to speculate that sequences important for the regulation of the expression of Bcin04g06240 and Bcin04g06250 are located within the DNA fragment deleted to generate the Δ*Bcfhg*1 strain, leading to the loss of regulated expression of both genes. 

Exposure to NO determined the induction of the expression of the nitrate and nitrite reductase genes, which were detected as differentially expressed in the *B. cinerea* wild-type strain. In fungi, the transcription of the genes encoding these two enzymes is strictly regulated as part of the complex global regulatory circuit that governs nitrogen metabolism. Nitrogen catabolite repression (NCR) is a broad program of transcriptional regulation that allows fungi to use alternative or non-preferential nitrogen sources (nitrate, nitrite, purines, amides, most amino acids and proteins) when preferential sources are not available or they are present in concentrations low enough to limit growth (ammonia, glutamine and glutamate) [67]. The use of any of these secondary nitrogen sources is regulated at the transcriptional level and almost always requires the synthesis of a set of pathway-specific catabolic enzymes and permeases that are otherwise subject to NCR. The synthesis of the enzymes of a particular catabolic pathway typically requires two distinct positive signals: a global signal indicating nitrogen derepression and a pathway-specific signal indicating the presence of a substrate or an intermediate of that pathway. The first signal is mediated by global trans-acting GATA factors, while the second by pathway-specific regulatory proteins [68]. In *Aspergillus* spp., a genus where NCR has been well characterized, the nitrate assimilation genes (nitrate reductase NIAD, nitrite reductase NIIA, a nitrate/nitrite transporter NRTA and a nitrite-specific transporter NITA) are clustered. A fifth gene, coding for another nitrate/nitrite transporter, was found to be separated on the same chromosome [68,69]. When the conditions are suitable—that is, there are low intracellular concentrations of the preferred nitrogen sources and the simultaneous presence of nitrate or nitrite as transcriptional inducers—the genes of this pathway are activated through the synergistic action of the general regulator of nitrogen metabolism AREA and the pathway-specific regulator NIRA [70,71]. Considerable experimental and genomic evidence suggests that this regulatory mechanism is conserved in the filamentous ascomycetes, although the clustering of the structural genes is not always conserved and the regulation of specific factors may vary among species [68,72,73]. The high level of expression of the *Bcnia*D and *Bcnii*A genes observed after 2 h exposure to NO of the B05.10 germinating spores indicated that nitrate accumulated intracellularly and that the gene products encoded by the orthologs of the regulatory genes areA and nirA in *B. cinerea*, *Bcare*A and *Bcnit*4, respectively, were functional and responded to the inducing signal. This nitrate, acting as a transcriptional inducer favoring the activation of the genes of its assimilation route, would result from the oxidation of NO by the action of the flavohemoglobin BCFHG1 and from the activity of the MNO acting on nitroalkanes [62]. In the experimental conditions utilized in this work, using a complex medium (PDB ½) that facilitates synchronous germination, the presence of traces of preferential or other non-preferential nitrogen sources that would affect the regulation of nitrogen utilization cannot be discarded. However, the transcriptional response of the *Bcnia*D and *Bcnii*A genes observed means that at the time point considered, the regulation of nitrogen source utilization favoring the assimilation of nitrate was operative and prevails, and NO could be considered as an alternative nitrogen source. 

The strong expression induction of *Bcnia*D and *Bcnii*A was mostly dependent on a functional flavohemoglobin. This observation supports a major role for the flavohemoglobin as the enzyme responsible for the generation of the signal needed to trigger their induction, nitrate. The much lower induction of both genes observed in the Δ*Bcfhg*1 strain (they are not identified as differentially expressed genes in this comparison) could be due to the transport of traces of NO_3_^−^ in the media into the fungal cells that accumulate to a larger extent when the Δ*Bcfhg*1 germinating spores are exposed to NO if the activity of the nitrate reductase is reduced by exposure to NO, as has been reported in *A. nidulans* [74]. The induction of *Bcnii*A could also be derived from the formation of nitrite, which could be produced spontaneously from NO oxidation in the medium and inside the cell [74,75]. This situation is contrary to that observed in *A. nidulans*, where treatment with a NO donor in both the wild-type strain and the Δ*fhbA*/Δ*fhbB* double mutant led to the same strong upregulation of the nitrate assimilation genes. The authors considered it unlikely that this induction was only a consequence of the formation of NO_3_^−^ by a flavohemoglobin [74]. 

The gene most highly induced in response to NO in both strains was the Bcin11g06310 gene. The encoded protein was annotated as a protein containing an NmrA-like domain protein. Two other genes encoding a protein with a predicted NmrA-like domain were also induced. Additionally, in the case of the Δ*Bcfhg*1 strain, seven additional genes encoding proteins containing NmrA-like domains were detected as DEGs. NmrA-like domain-containing proteins are a conserved group of transcriptional modulators that translate metabolic fluctuations to changes in gene expression through interaction with transcription factors. These transcriptional modulators sense the metabolic state of the cell by binding specifically to a cofactor, such us NAD(P)(H), and this binding modulates their ability to interact with other proteins [76,77]. The *A. nidulans* NmrA protein belongs to this group. It has been shown to bind NAD(P)+ and to mediate NCR [78]. Under nitrogen sufficiency conditions, *nmr*A expression is high [72], and the protein binds to the AreA protein, reducing its activity [72,77,79,80] and thus preventing nitrogen catabolic genes’ expression. None of the genes encoding the NmrA-like domain-containing proteins whose expression was induced in response to NO were the ortholog of the *A. nidulans nmr*A gene (in the *B. cinerea* genome, this was predicted to be Bcin15g02310—data not shown). Therefore, no effect on the transcription of this regulatory factor by the exposure to NO that could affect nitrogen metabolism genes can be inferred from our results. However, the number of NmrA-like domain-containing proteins that were induced is noteworthy. Since these proteins have been proposed to be a direct link to translate changes in cellular metabolism into transcriptional responses, through the diffusible metabolic cofactors [76], this observation could be indicative of a severe alteration of the cellular status—particularly of the redox status—which might determine changes in transcription patterns that would result in alterations in primary and secondary metabolism.

An affectation of the expression of secondary metabolism genes by NO was observed both in the wild-type and mutant strains but was accentuated in the latter. Our transcriptomic analysis, therefore, supports recent data showing the involvement of NO in modulating fungal secondary metabolism [9]. Remarkably, the participation of NmrA-like proteins in the regulation of aflatoxin and bikaverin biosynthesis in *A. flavus* and *F. fujikuroi*, respectively, has been observed [81,82].

In the wild-type strain, the transcriptional response was less extensive, supporting the protective role of the flavohemoglobin. This response, under the criteria considered in this analysis to identify DEGs, involved changes in the expression of only one transcription factor coding gene, Bcin05g04650, the ortholog of the *A. nidulans abaA* gene. The activity of BCFHG1 likely keeps the levels of intracellular NO low, limiting its effects, which are mainly derived from transient direct modifications of proteins. In the deprotected system, the production and accumulation of reactive oxygen and nitrogen forms likely generate a long-lasting status of redox stress, which drives several physiological responses and involves the regulation of several transcription factors. Eleven transcription factors of different types were differentially expressed—nine upregulated and two downregulated—in the Δ*Bcfhg*1 strain, predominating the Zn2C6 class, which constitutes the most numerous class of transcription factors in *B. cinerea* [83]. It will be interesting to determine the general or specific functions of these transcription factors. They represent candidates of transcription factors specifically mediating the response to nitrosative stress conditions in *B. cinerea*. To date, no functional characterization of any of them has been given. Interestingly, the most highly induced putative transcription factor (log_2_FC = 6.76) is a member of the C2H2 class.

The specific induction in response to NO of Bcin05g04650 (*Bcaba*A) in the wild-type strain is intriguing. *abaA* is a master regulator of asexual development in filamentous fungi [84]. In *A. nidulans*, it governs, together with *Brl*A and *Wet*A, the gene regulatory network that controls the development of the asexual fruiting body, regulating specifically the development of the phialides [85]. This role is conserved in different fungi. In our study, the phenotype of the wild-type spores incubated in the presence of NO resembled that observed in strains of *F. graminearum* overexpressing *abaA*, which manifested, among other pleiotropic defects, a decrease in the germination rate of the mutants at 4 and 6 h after incubation. However, there was no significant difference in the germination rates between wild-type and *abaA*-overexpressing strains after incubation for 12 h. This observation indicates that additional functions of *abaA* exist in some filamentous fungi besides its participation in conidiation. This, together with reports that *abaA* is involved in the positive regulation of nuclear division in *A. nidulans* and *Penicillium marneffei* [86,87] and in the control of maturation and dormancy in *F. graminearum* conidia through cell cycle regulation [88], suggests the involvement of this transcription factor in the delayed germination of wild-type spores observed in our study.

NO exposure also had an impact on the expression of nuclear genes encoding factors involved in mitochondrial metabolism. At low concentrations, for short periods, NO specifically and reversibly inhibited cytochrome oxidase. At higher levels, through prolonged exposure to NO or the generation of other more reactive oxides, particularly peroxynitrite by reacting with superoxide, this determines the irreversible inhibition of respiration and other damage in mitochondria [89]. The upregulation of *Bccox*17 (Bcin02g03350), Bcin07g05710 and *Bcnde*4 (Bcin03g08570) suggests a limited functionality of cytochrome c oxidase in the electron transport chain and an increase in the alternative respiration pathway in the mutant spores. This could be the result of the reprogramming of the mitochondria through retrograde signaling due to the altered redox state of its cells [90]. It is interesting to note that the gene encoding the *B. cinerea* alternative oxidase was also upregulated in the Δ*Bcfhg*1 strain, although it did not reach the values to be considered a DEG (data not shown). On the other hand, the expression pattern of the glutamate decarboxylase enzymes encoded by *Bcgad*1 (Bcin09g05020), specifically downregulated in the wild-type strain, and by Bcin03g01040, specifically downregulated in the Δ*Bcfhg*1 strain, could indicate a possible impact on the GABA shunt pathway [45].

The downregulation of *Bcgph*1 (Bcin15g03620), encoding a glycogen phosphorylase, and *Bcgdb*1 (Bcin01g10310), encoding the glycogen debranching enzyme, in the Δ*Bcfhg*1 strain, which are both enzymes involved in glycogen mobilization, is remarkable. As in most organisms, glycogen constitutes a key storage compound in fungi. In *M. oryzae*, it has been shown that glycogen reserves in the spore are broken down during germination and that the rapid degradation of lipid and glycogen reserves contributes to the generation of the turgor necessary for appressorium formation [91,92]. Glycerol metabolism was also affected, as the glycerol kinase coding gene (Bcin09g04020) and the glycerol 2-dehydrogenase coding gene (Bcin11g04710) were downregulated. Therefore, several primary metabolism pathways involved in energy production appeared to be downregulated in Δ*Bcfhg*1, which could condition early development and explain, at least partially, the blockage in the germination process observed in the mutant strain.

Several genes related to the terms glucan and chitin appeared in the lists of DEGs, which is likely a reflection of alterations in the cell wall remodeling processes that take place during the germination and polarized growth of germ tubes and that are affected by exposure to NO. *Bccda*1 (Bcin03g05710) is one of these genes. It codes for a chitin deacetylase, an enzyme that carries out the deacetylation of chitin, forming chitosan, and appeared to be induced in both strains in our analysis. Its upregulation was strictly correlated with germination, independent of the germination conditions, and was positively regulated by *Bmp*1 MAP kinase cascade [93]. It was rapidly and very strongly induced within the first hour of spore incubation and then declined slowly, maintaining notable levels of expression at 4 and 15 h. It is intriguing why, upon exposure to NO, the expression of *Bccda*1 was stimulated when it appears that germination was being slowed down or completely blocked.

The calcineurin pathway could also have been affected by NO. Calcineurin and cyclophilin A are both cellular components involved in fungal morphogenesis and virulence. The impairment of calcineurin signaling in fungi results in pleiotropic phenotypes. In *A. fumigatus*, calcineurin mutants displayed severe defects in conidial germination, polarized hyphal growth and conidium development [94,95,96]. In *B. cinerea*, calcineurin and cyclophilin A have been shown to be involved in different aspects of morphogenesis and development [97]. The mutant for the calcineurin catalytic subunit *BccnA* showed a severe growth defect, did not produce conidia and was avirulent [98]. The disruption of the cyclophilin gene *Bccyp*2, an inhibitor of calcineurin, did not impair the pathogen mycelial growth, osmotic and oxidative stress adaptation or cell wall integrity but delayed conidial germination and germling development; altered conidial and sclerotial morphology and reduced infection cushion formation, sclerotial production and virulence [99]. The transcriptional changes in the cyclophilin-dependent genes *Bccpd*3 (Bcin01g01590), downregulated in the wild-type strain, and *Bccpd*1 (Bcin07g07020), upregulated in the mutant strain, and in the calcineurin-dependent gene *Bcgas*2 (Bcin14g04260), downregulated in both strains, suggest that NO could perform signaling whilst making use of this pathway.

In summary, our results show that in *B. cinerea*, NO metabolism affects germination and is connected to the nitrate assimilation pathway, these effects being dependent on the flavohemoglobin BCFHG1. The comparative transcriptomic analysis carried out in response to NO exposure widens our knowledge on the genetic basis of the physiological responses activated under conditions of nitrosative stress in filamentous fungi. 

## Figures and Tables

**Figure 1 jof-08-00699-f001:**
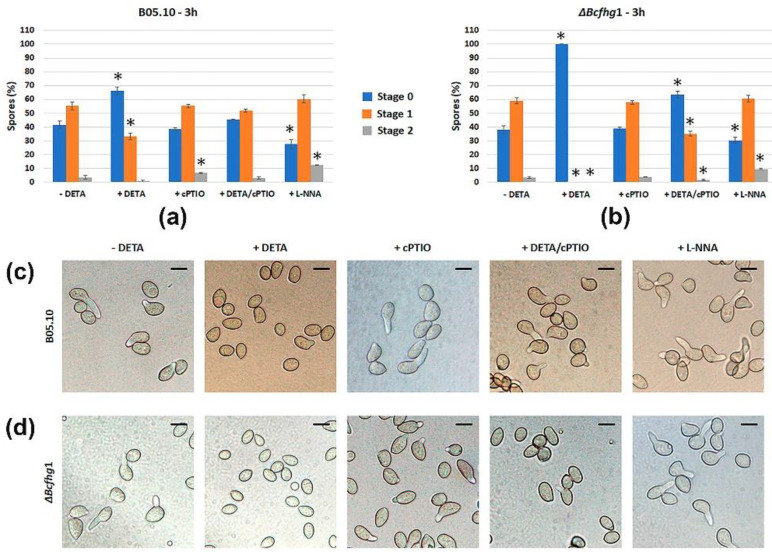
Effect of addition of the NO donor DETA, the NO scavenger cPTIO and the NOS inhibitor L-NNA on the germination of spores of the *B. cinerea* strains B05.10 (**a**) and Δ*Bcfhg*1 (**b**) after 3 h of incubation. The reagent concentrations were 250 μM DETA, 500 μM cPTIO, and 1 mM L-NNA. The graphs show the means of three replicates ± SD (*n* > 100 each). (* indicates significant differences at *p* < 0.05—Student’s *t*-test—between each stage and treatment and its control stage in the absence of DETA.) (**c**,**d**) Representative images of the germination process of each strain at the same time point and treatments as in (**a**,**b**). Scale bars, 10 μm.

**Figure 2 jof-08-00699-f002:**
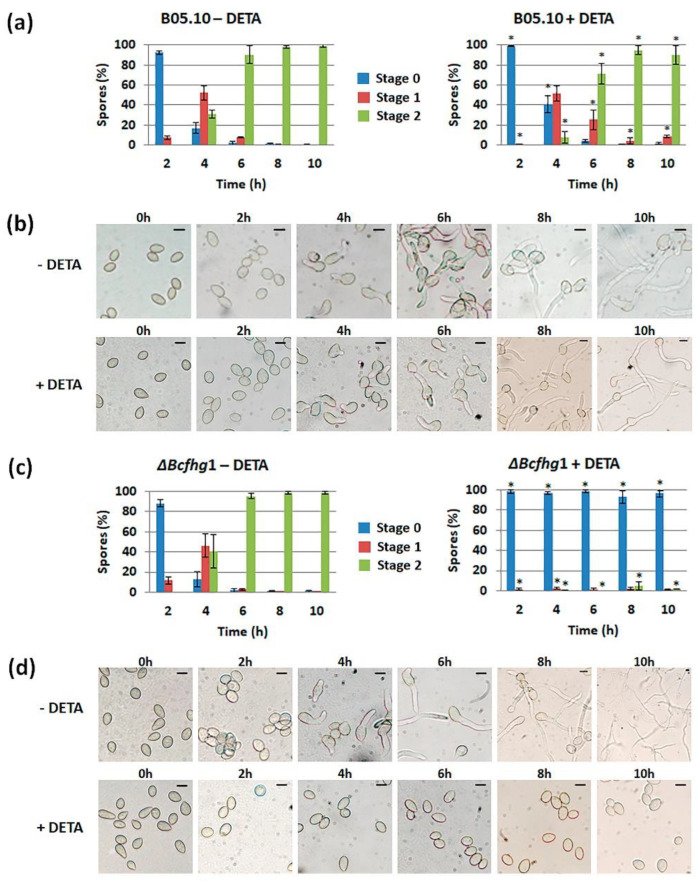
Germination kinetics of *B. cinerea* B05.10 (**a**) and Δ*Bcfhg*1 (**c**) spores cultured in PDB ½ medium under static culture conditions. Germination was evaluated by quantifying the number of spores in stages 0, 1 or 2 at the indicated time points. The graphs show the means of three replicates ± SD (*n* > 100 each). (* indicates significant differences at *p* < 0.05—Student’s *t*-test—in the presence of 250 μM DETA and its reference situation in the absence of DETA.) (**b**–**d**) Representative images of the germination process of each strain at the time points considered. Scale bars, 10 μm.

**Figure 3 jof-08-00699-f003:**
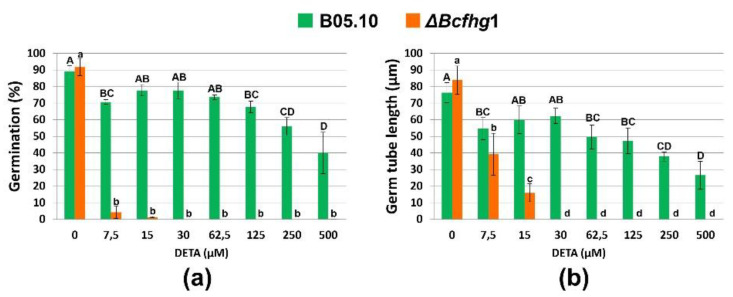
Effects of DETA on germination (**a**) and germ tube growth (**b**) in *B. cinerea* strains B05.10 and Δ*Bcfhg*1. Percentage of germinated spores (stage 1 + stage 2) and germ tube length were determined after 4 and 10 h of incubation, respectively, under the different concentrations of the NO donor indicated. Values are the means of three replicates ± SD (*n* > 100 each). Differences between concentrations for each strain were tested by one-way ANOVA followed by the Bonferroni test at *p* < 0.05. Uppercase letters represent groups of the B05.10 strain. Lowercase letters represent groups of the Δ*Bcfhg*1 strain.

**Figure 4 jof-08-00699-f004:**
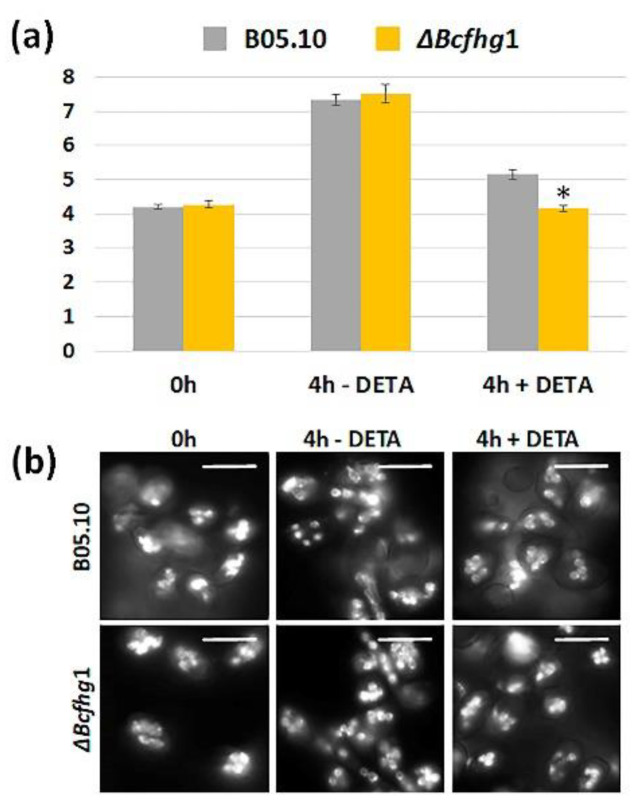
Effect of NO on nuclear division in *B. cinerea* germinating spores. (**a**) Histogram showing the average number of nuclei in conidia of the B05.10 and Δ*Bcfhg*1 strains incubated in static culture in PDB 1/2 during 4 h in the absence and in the presence of 250 µM DETA. The data present the means of three replicates ± SD (*n* ≥ 100 each). (* indicates significant differences at *p* < 0.05—Student’s *t*-test—in the presence of DETA in the Δ*Bcfhg*1 strain in comparison with the B05.10 strain.) (**b**) Representative images of DAPI staining preparations of spores of the B05.10 strain and the Δ*Bcfhg*1 mutant strain at the time points and conditions considered in (**a**). As the reference, spores of each strain collected at 0 h were analyzed. Scale bars, 10 µm.

**Figure 5 jof-08-00699-f005:**
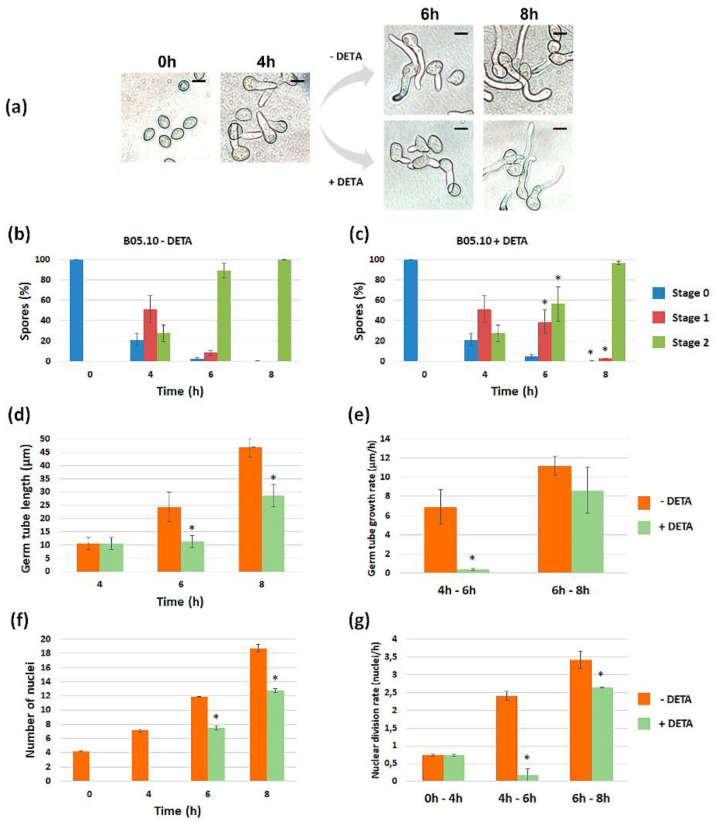
Effect of the addition of 250 μM DETA to B05.10 spores once the germination program had been launched. (**a**) Experimental set-up and representative images taken at the indicated time points and conditions. (**b**,**c**) Percentages of spores in the different stages of development (0, 1 and 2) at the indicated time points in the absence (**b**) and presence (**c**) of 250 μM DETA. (**d**) Average germ tube length at the indicated time points. (**e**) Germ tube growth rate at 4–6 and 6–8 h. (**f**) Average number of nuclei in spores and germ tubes at the indicated time points. (**g**) Average nuclear division rate at 4–6 and 6–8 h. The data present the means of three replicates ± SD (*n* ≥ 100 each). (* indicates significant differences at *p* < 0.05—Student’s *t*-test—in the presence of 250 μM DETA and its reference situation in the absence of DETA.) Scale bars, 10 µm.

**Figure 6 jof-08-00699-f006:**
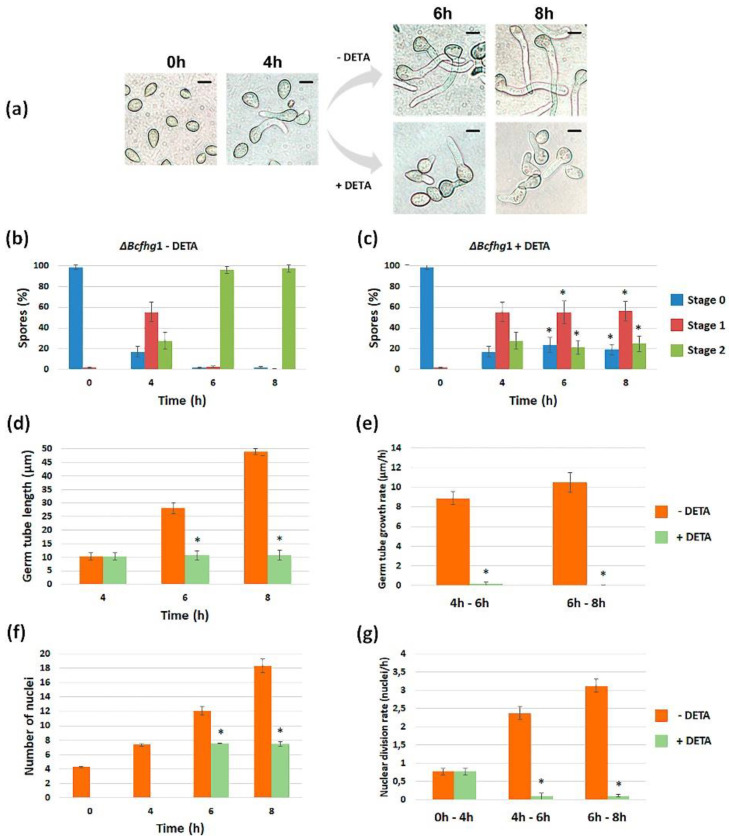
Effect of the addition of 250 μM DETA to Δ*Bcfhg*1 spores once the germination program has been launched. (**a**) Experimental set-up and representative images taken at indicated time points and conditions. (**b**,**c**) Percentages of spores in the different stages of development (0, 1 and 2) at the indicated time points in the absence (**b**) and presence (**c**) of 250 μM DETA. (**d**) Average germ tube length at the indicated time points. (**e**) Germ tube growth rate at 4–6 and 6–8 h. (**f**) Average number of nuclei in spores and germ tubes at the indicated time points. (**g**) Average nuclear division rate at 4–6 and 6–8 h. The data present the means of three replicates ± SD (*n* ≥ 100 each). (* indicates significant differences at *p* < 0.05—Student’s *t*-test—in the presence of 250 μM DETA and its reference situation in the absence of DETA.) Scale bars, 10 µm.

**Figure 7 jof-08-00699-f007:**
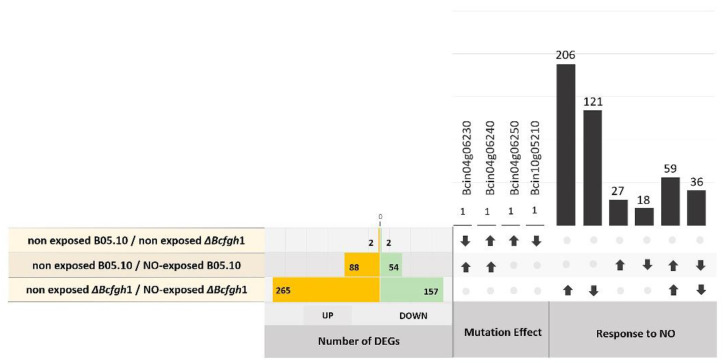
Summary of the DEGs detected in the differential gene expression analysis in response to NO in the B05.10 and Δ*Bcfhg*1 strains.

**Figure 8 jof-08-00699-f008:**
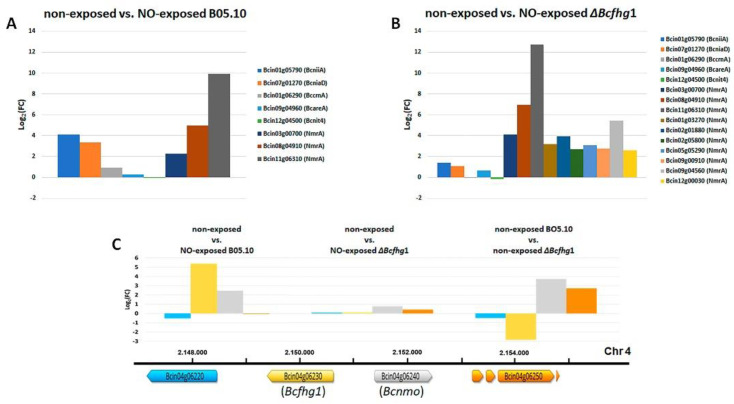
Changes in the expression levels of genes related to nitrate assimilation and of NmrA-like domain-containing protein encoding genes in the comparison of non-exposed and NO-exposed B05.10 (**A**) and non-exposed and NO-exposed Δ*Bcfhg*1 (**B**). (**C**) Change in the expression levels of the genes Bcin04g06220, Bcin04g06230 (*Bcfhg*1), Bcin04g06240 (*Bcnmo*) and Bcin04g06250 in the comparison of non-exposed and NO-exposed B05.10, non-exposed and NO-exposed Δ*Bcfhg*1 and non-exposed BO5.10 and non-exposed Δ*Bcfhg*1. Below the histograms, a representation of the loci they occupy is shown.

## Data Availability

All RNA-seq data are deposited in the National Center for Biotechnology Information (NCBI) Sequence Read Archive (SRA) (https://www.ncbi.nlm.nih.gov/sra/) under the BioProject PRJNA851784. Accession numbers for datasets are as follows: KO_4, SAMN29251512; KO_5, SAMN29251513; KO_6, SAMN29251514; KO_7, SAMN29251515; KO_9, SAMN29251516; WT_4, SAMN29251517; WT_5, SAMN29251518; WT_6, SAMN29251519; WT_7, SAMN29251520; WT_8, SAMN29251521; WT_9, SAMN29251522.

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
