# Peer review of "Nitric Oxide Metabolism Affects Germination in Botrytis cinerea and Is Connected to Nitrate Assimilation"

_jof, 2022, doi:10.3390/jof8070699_

Round 1

Reviewer 1 Report

The article “Nitric oxide metabolism affects germination in Botrytis cinerea and is connected to nitrate assimilation” by Fernández et al., is very interesting and insightful for the NO research community. I really enjoy reading the article. Following are my specific observations:  

Section 2.2 Most of the spore germination experiments were performed in the ½ PDB which is full of nutrients and supports germination and thus do not reflect the true phenotypes in cases when we use some additional inhibitors or substrates. Did authors perform these experiments in minimal media Czapek Dox or glucose phosphate with or without different NO modulator or inhibitors? While agitation provide complete mixing and continuous supply of added inhibitors, why authors avoid shaking or agitation of the cultures while spore germination and choose static conditions?

Section 2.6 For differential gene expression analysis the culture was incubated for total 6 hours. 4 hours to initiate the germination + 2 hours after adding DETA, creating the time lag of collecting the samples. Did authors also collect similar samples for WT and mutant? Please explain clearly in the text and mention the addition of DETA to WT also.

Section 3.1 Please show that the expression of Bcfhg1 is reduced in the ΔBcfhg1 mutants compared to WT BO5.10. This data can go in Supplementary.

Since authors have multiple time point and treatments, statistical analysis should be performed with ANNOVA and not with T-test. How T-test is done, * appears over bars are bit confusing. It is not clear how comparisons are made, For example- Fig-1 panel (a and b), on the treatment of +L-NNA in WT and  ΔBcfhg1 both contains * at stage 0 and 2, whereas +DETA/cPTIO do not contains. If I am not wrong comparison are made between each stage of WT vs mutant, in such case +DETA/cPTIO should also be significantly different.  Please perform ANNOVA. You can also think of making stack graph for data having germination stages.  Similar is the case in Fig 2, 5 and 6.

Fig. 31 should be Fig 3. 

Author Response

The article “Nitric oxide metabolism affects germination in Botrytis cinerea and is connected to nitrate assimilation” by Fernández et al., is very interesting and insightful for the NO research community. I really enjoy reading the article. Following are my specific observations:  

Section 2.2 Most of the spore germination experiments were performed in the ½ PDB which is full of nutrients and supports germination and thus do not reflect the true phenotypes in cases when we use some additional inhibitors or substrates. Did authors perform these experiments in minimal media Czapek Dox or glucose phosphate with or without different NO modulator or inhibitors? While agitation provide complete mixing and continuous supply of added inhibitors, why authors avoid shaking or agitation of the cultures while spore germination and choose static conditions?

We did not perform these experiments in minimal media. In some preliminary experiments we found that germination in those media were not very efficient and synchronous, while in the conditions finally used in this work most of the spores germinated, did it within a narrow time window and the effects of the chemicals were reproducible and scorable. Furthermore, and importantly, we avoided shaking because when cultured in agitation we observed aggregation of the germinating spores and germlings, making it difficult to evaluate germination stages and germ tube length. Without shaking we could make these estimations more precisely.

Section 2.6 For differential gene expression analysis the culture was incubated for total 6 hours. 4 hours to initiate the germination + 2 hours after adding DETA, creating the time lag of collecting the samples. Did authors also collect similar samples for WT and mutant? Please explain clearly in the text and mention the addition of DETA to WT also.

Yes, the spores of both the wild-type strain and of the ∆Bcfgh1 mutant strain were collected in parallel in the same conditions and DETA was added to both samples in the same conditions. This is now clearly indicated in the text in section 2.6. 

Section 3.1 Please show that the expression of Bcfhg1 is reduced in the ΔBcfhg1 mutants compared to WT BO5.10. This data can go in Supplementary.

We apologize, but we don’t understand well this indication. In section 3.1. we focus on the pharmacological analysis, and we think we don’t make a comment on the expression of Bcfhg1 in the ∆Bcfgh1 mutant strain. In section 3.2.2., before describing the analysis of differential gene expression in the mutant strain -DETA vs. +DETA we considered the comparison between the wild type strain and the ∆Bcfgh1 when growth conditions in the absence of DETA are considered. And it is in this comparison in which the expression of Bcfhg1 is reduced in the mutant in comparison with the wild type, as expected since most of the coding region of the Bcfhg1 gene has been deleted by gene replacement. This downregulation of Bcfhg1 in the mutant strain in comparison with the wt strain at 6h in the absence of DETA is offered in Table S4.

Since authors have multiple time point and treatments, statistical analysis should be performed with ANNOVA and not with T-test. How T-test is done, * appears over bars are bit confusing. It is not clear how comparisons are made, For example- Fig-1 panel (a and b), on the treatment of +L-NNA in WT and  ΔBcfhg1 both contains * at stage 0 and 2, whereas +DETA/cPTIO do not contains. If I am not wrong comparison are made between each stage of WT vs mutant, in such case +DETA/cPTIO should also be significantly different.  Please perform ANNOVA. You can also think of making stack graph for data having germination stages.  Similar is the case in Fig 2, 5 and 6.

We are always considering two series of data, most of the times one generated in the presence of DETA and one generated in the reference situation, the absence of DETA (or of any chemical, just the growth medium). In Figure 1 in panel (a) the comparison is done for the wild-type strain for each stage (0, 1 or 2) in the presence of a given chemical and the absence of any chemical (just the growth medium). In panel (B) the same analysis is performed, but with the mutant strain. The analysis is not performed between each stage of WT and mutant strains. We have tried to make this clearer being more descriptive in the figure legend. And the * on top of a bar indicates significant differences in that situation (DETA, or in this figure the different chemicals considered) at a given stage, 0, 1 or 2, in comparison with the reference, that is, that stage in the absence of DETA (-DETA). In Figures 2, 5 and 6 we are also comparing for each strain and time point two series of data, with and without DETA. In these figures we have different time points, and the effect of DETA at each time point for each stage is analyzed again in comparison with the corresponding reference sample, without DETA.

We have included the statistical analysis of Figure 3, which in this case is performed using a one-way ANOVA analysis as now we are analyzing simultaneously several series of data. It is described in the Figure legend. 

Fig. 31 should be Fig 3. 

In the File we have and uploaded we can’t see this problem. In the revised version we are uploading now the numbering of the figures is correct.  

Thank you very much for your comments and suggestions

Reviewer 2 Report

The authors demonstrated that NO inhibits germination of Botrytis cinerea spore. And also, the transcriptomic analyses suggest several possible pathway to regulate germination induced by NO. This manuscript provides the new function of NO in germination in B. cinerea. However, the manuscript has many point to be improved as described below.

Major points

1.      Figure 1. Why did the addition of cPTIO slightly promote germination? Did cPTIO scavenge NO synthesized endogenously? I recommend the authors to analyze the intracellular NO level without NO exposure.

2.      Figure 1. Did L-NNA really inhibit endogenous NO production in this fungi? Does this fungi conserve an orthologue of mammalian NOS? Or did the authors measure the intracellular NO content? L-NNA is an analogue of arginine, thus it is possible this compound exerts it function by disturbing arginine metabolism.

3.      In abstract, the authors mentioned the possibility that nitrogen assimilation system including BCFHG1 and NMO function for signal transduction, which seems one of the conclusions of this study. However, no conclusions related to BCFHG1 and NMO were described in the results section (3.2. DEGs in response to NO). No clear conclusions are presented in this section 3.2. Please make some conclusions from DEGs analyses.

4.      The manuscript is totally tedious especially in the results section 3.2. (DEGs in response to NO). The main topic of this manuscript seems that NO functions in germination are related to nitrogen assimilation, from its title. However, the most parts of result sections are just lists of DEGs in each condition, even though these are not directly required for the conclusions. The detail information of DEGs analyses, by which the conclusions were not made in relation with the research purpose, NO functions in development (could be germination process) of B. cinerea (L113-114), should be included in discussion section with the supplementary information.

5.      In the final paragraph in discussion section (L972-974), the authors mentioned that NO metabolism affects germination and is connected to nitrate assimilation pathway. The connection between NO metabolism and nitrate assimilation seems a well-known phenomenon, as the authors cited the paper of A. nidulans (L851-854) and speculated that nitrate, an enzymatic reaction product of flavohemoglobin can be an inducer of nitrate assimilation system. If the main conclusions of this manuscript are the relationship between NO and germination in this fungi and the connection of NO metabolism with nitrate assimilation, its impact is very low.

Or if the nitrate assimilation is linked to the regulation of germination by NO, it is one of good conclusion with higher significance. To mention this connection between germination inhibitory effects of NO with nitrate assimilation, the additional experiments should be performed. For example, the germination rate should be analyzed using the strain lacking the genes encoding nitrate reductase and/or nitrite reductase. Or does nitrate function as an alternative inhibitor of germination instead of NO?

Minor points

1.      The manuscript totally contains many points to be corrected in terms of English, thus it is hard to read and understand accurately. Please revise the manuscript with the aid of English proofreading service.

2.      L108. The font of NO3- is different from other parts.

3.      L144. Does DETA means DETA NONOate? Please indicate the formal name of this NO donor.

4.      L198-204. In the sample preparation for RNA sequencing, the control sample was prepared by the continuous culture for 6 h without any manipulation corresponding to the NO treatment in the preparation of NO-treated sample. If the spores were collected by centrifuge to 50 mL tube, the centrifuge process might affect gene expression? Was the spore collected by centrifuge?

5.      L382-385. This experiment (Figure 5) could clarify what stages of germination process is affected by NO, though the manuscript shows to find the degree of immediacy of NO function.

6.      In the manuscript, many gene names were present thus it is hard to follow the logic flow and story. I strongly recommend the authors to modify the manuscript, figures, and figure legends to make it easy to follow. For example, the sample labels should be accompanied with gene product name like NmrA in Figure 8a and 8b.

7.      I recommend the authors to prepare the supplementary table(s) to summarize the possible pathways of NO to regulate germination based on the results of DEGs analyses, for easy understanding of your proposed pathway described in discussion section.

8.      Some figures lack statistical analyses, such as Fig.3 and 4.

Author Response

The authors demonstrated that NO inhibits germination of Botrytis cinerea spore. And also, the transcriptomic analyses suggest several possible pathway to regulate germination induced by NO. This manuscript provides the new function of NO in germination in B. cinerea. However, the manuscript has many point to be improved as described below.

 Major points

  1. Figure 1. Why did the addition of cPTIO slightly promote germination? Did cPTIO scavenge NO synthesized endogenously? I recommend the authors to analyze the intracellular NO level without NO exposure.

Yes, we agree with the reviewer. We also attribute this slight effect promoting germination to cPTIO scavenging endogenously synthetized NO. A comment has been added in the text regarding this observation in section 3.1.1.

We did describe the detection of endogenous production of NO in B. cinerea in a previous work using fluorescent probes and we also described that cPTIO scavenged the NO produced (ref. 36 in the manuscript). In the work submitted now we have not quantified the level of endogenously produced NO but we think that, from those previous analysis, we can accept that in the wild type strain, cPTIO scavenges the NO being produced. But we understand the reviewer’s recommendation. As this work is part of a long-term ongoing research, we think there are several aspects being raised that deserve further consideration and we suggest that this point can be investigated in the near future. Quantification of endogenous production in different developmental stages and in different media, and in different mutants (nitrate reductase and nitrite reductase mutants) can provide further insight into the characterization of the system.  

  1. Figure 1. Did L-NNA really inhibit endogenous NO production in this fungi? Does this fungi conserve an orthologue of mammalian NOS? Or did the authors measure the intracellular NO content? L-NNA is an analogue of arginine, thus it is possible this compound exerts it function by disturbing arginine metabolism.

We made the interpretation that the slight increase in germination can be considered an indirect indication of a reduction of endogenous NO levels upon treatment with L-NNA. Following literature, we accepted that L-NNA inhibits nitric oxide synthases found in eukaryotes, but we can’t assure that L-NNA is inhibiting endogenous NO production in B. cinerea as we have not quantified the levels of intracellular NO content. This is something that we can also consider in our future work. For that reason, we stated just that these results “suggest” that a fraction of the NO produced by the fungus could be derived from enzymes related to mammalian nitric oxide synthases. An orthologue of genes encoding a nitric oxide synthase in the B. cinerea genome has not been identified and we have discussed this question in our group trying to figure out possible explanations. In an interesting work by Sarkar et al. (2014) (PLoS ONE 9(9): e107348. doi:10.1371/journal.pone.0107348) in Macrophomina phaseolina, also a necrotroph, the authors indicated that the two domains found in NO synthases, the oxygenase domain and the flavodoxin/nitric oxide synthase domain, were encoded by separate ORFs in this fungus and that NO production could be explained even in that situation. Perhaps the analysis of the organization of the domains of this enzyme has to be revisited in Botrytis and in other fungi. If this is also the case in Botrytis, and being the domains functional, L-NNA could exert its function as an inhibitor to some extent. The possibility that L-NNA could disturb arginine metabolism in Botrytis can’t be discarded and it is something we have not tested. We assume, following the literature and the analysis of the effect of NO in germination in Colletotrichum coccodes using D-NNA and L-NNA (ref. 13 in the list of references) that the accelerated germination caused by L-NNA was not a nutritional effect. Although we have these considerations in mind, the observation and discussion of the effect of L-NNA was not the main objective of our work. We just wanted to mention it in the discussion because we considered it adds on previous observations in other systems.

  1. In abstract, the authors mentioned the possibility that nitrogen assimilation system including BCFHG1 and NMO function for signal transduction, which seems one of the conclusions of this study. However, no conclusions related to BCFHG1 and NMO were described in the results section (3.2. DEGs in response to NO). No clear conclusions are presented in this section 3.2. Please make some conclusions from DEGs analyses.

The main message we want to transmit is that NO affects germination and that a connection with nitrate assimilation metabolism is found. BCFHG1, and NMO as it also generates nitrate, is shown to be responsible for the generation of the signal triggering the induction of the nitrate genes. Honestly, we did not want to pose a more general function on signal transduction for the nitrogen assimilation genes, and in particular for the BCFHG1 and NMO coding genes. We have added comments in section 3.2.1., as a conclusion, highlighting the fact that nitrate, the signal needed to trigger the expression of the nitrate and nitrate reductase genes, which appear upregulated in this comparison, is likely generated by the activity of these two NO detoxifying systems. 

  1. The manuscript is totally tedious especially in the results section 3.2. (DEGs in response to NO). The main topic of this manuscript seems that NO functions in germination are related to nitrogen assimilation, from its title. However, the most parts of result sections are just lists of DEGs in each condition, even though these are not directly required for the conclusions. The detail information of DEGs analyses, by which the conclusions were not made in relation with the research purpose, NO functions in development (could be germination process) of B. cinerea (L113-114), should be included in discussion section with the supplementary information.

We have extensively reformulated and reduced sections 3.2.1. and 3.2.2. We have focused essentially on the genes detected in the analysis which inform about processes which appear to be related to the effect of NO on germination and on the response of the physiology of the fungus to NO. In this formulation, short conclusions are presented which are the basis for the discussion section. Two sheets, one in Table S2 and one in Table S3, have been included presenting a summary of the genes and processes which are referred to in the text in sections 3.2.1. and 3.2.2. and considered in the discussion section.   

  1. In the final paragraph in discussion section (L972-974), the authors mentioned that NO metabolism affects germination and is connected to nitrate assimilation pathway. The connection between NO metabolism and nitrate assimilation seems a well-known phenomenon, as the authors cited the paper of A. nidulans (L851-854) and speculated that nitrate, an enzymatic reaction product of flavohemoglobin can be an inducer of nitrate assimilation system. If the main conclusions of this manuscript are the relationship between NO and germination in this fungi and the connection of NO metabolism with nitrate assimilation, its impact is very low.

Or if the nitrate assimilation is linked to the regulation of germination by NO, it is one of good conclusion with higher significance. To mention this connection between germination inhibitory effects of NO with nitrate assimilation, the additional experiments should be performed. For example, the germination rate should be analyzed using the strain lacking the genes encoding nitrate reductase and/or nitrite reductase. Or does nitrate function as an alternative inhibitor of germination instead of NO?

The reviewer raises an important point. In the manuscript we are posing the conclusion that NO metabolism affects germination and that its detoxification is connected to nitrate assimilation. The connection with nitrate assimilation has been described in the work performed in A. nidulans, a very nice and mind opening work, but as far as we are aware there are not many other reports of that connection in fungi. Our analysis in B. cinerea supports those observations and we humbly consider that it represents a useful addition to literature, in particular if we keep in mind that differences between the two fungi are observed, as in both, exposition to exogenous NO causes induction of nitrate assimilation genes, but in B. cinerea the induction appears to be dependent on the flavohemoglobin while in A. nidulans, in the double mutant in the two flavohemoglobins described in this fungus, the induction of nitrate assimilation genes was maintained.

We agree that if it could be demonstrated that the assimilation of nitrate is linked to the regulation of germination that would be more meaningful. We think that the observations presented in our manuscript add on other observations, as indicated in the discussion, regarding the expression pattern of the flavohemoglobin during germination and elongation of germ tube, and all taken together suggest that that connection exists. And that is the point we would like to be taken from our observations. The reviewer indicates that additional experiments should be performed in order to gain information in this context, such as the evaluation of germination in nitrate reductase mutants. This kind of experimental approach is being taken into consideration and we already asked for the mutants to our collaborators. As the analysis of NO functions in Botrytis physiology is a long-term research project in our group, such an analysis will be incorporated in our plans. We apologize, but we have not tested in the course of this work if nitrate functions as an inhibitor of germination in B. cinerea.

Minor points

  1. The manuscript totally contains many points to be corrected in terms of English, thus it is hard to read and understand accurately. Please revise the manuscript with the aid of English proofreading service.

Following the recommendation, the manuscript has been revised by the English proofreading service offered by the Editorial.

  1. L108. The font of NO3- is different from other parts.

This has been corrected.

  1. L144. Does DETA means DETA NONOate? Please indicate the formal name of this NO donor.

Yes, it does. The formal name is now indicated in this section of Materials and Methods. Throughout the manuscript, DETA is maintained for simplicity. 

  1. L198-204. In the sample preparation for RNA sequencing, the control sample was prepared by the continuous culture for 6 h without any manipulation corresponding to the NO treatment in the preparation of NO-treated sample. If the spores were collected by centrifuge to 50 mL tube, the centrifuge process might affect gene expression? Was the spore collected by centrifuge?

The spores’ suspensions were just transferred to a 50 mL tube and DETA was then added to the tube to facilitate homogenous distribution of the chemical and uniform exposure of the spores. The mixture was immediately returned to the plate. This took very little time and the spores were not sedimented by centrifugation at this stage. We considered this manipulation was not causing changes in gene expression in our experimental set up, in particular if we keep in mind that incubation was then extended for two additional hours.

  1. L382-385. This experiment (Figure 5) could clarify what stages of germination process is affected by NO, though the manuscript shows to find the degree of immediacy of NO function.

This is an interesting observation we have also considered and discussed internally. But we must say that the experiment was not conceived to evaluate this question (at that moment we were addressing the problem of the evaluation and quantification of the effect of NO and the importance of the role of the flavohemoglobin), although the information generated can provide indications of early stages or processes being affected. For that to be evaluated, perhaps it would have been more informative making the same analysis adding the exogenous NO also earlier (may be after 2 h of incubation) and later (6 h of incubation). But it was not considered at that moment. And with the data collected, we think we can’t make a strong statement about it.

  1. In the manuscript, many gene names were present thus it is hard to follow the logic flow and story. I strongly recommend the authors to modify the manuscript, figures, and figure legends to make it easy to follow. For example, the sample labels should be accompanied with gene product name like NmrA in Figure 8a and 8b.

We have modified the manuscript trying to follow the recommendation. In this sense, we have reduced and simplified the descriptions in sections 3.2.1. and 3.2.2. and hope the logic can be followed now more easily. Figure 8a and 8b has been modified as suggested.    

  1. I recommend the authors to prepare the supplementary table(s) to summarize the possible pathways of NO to regulate germination based on the results of DEGs analyses, for easy understanding of your proposed pathway described in discussion section.

A new sheet has been added in Tables S2 and S3. In each case, relevant genes are presented organized following the groups mentioned in the description of processes that have been found as affected by exposition to NO and which are considered in the discussion section. We hope this help to follow the message offered in the manuscript. 

  1. Some figures lack statistical analyses, such as Fig.3 and 4.

The statistical analysis has been done and included in both figures.

Thank you very much for your comments and suggestions